# Bioinspired nondissipative mechanical energy storage and release in hydrogels via hierarchical sequentially swollen stretched chains

Henri Savolainen, Negar Hosseiniyan, Mario Piedrahita-Bello ✉ & Olli Ikkala ✉

Nature suggests concepts for materials with efficient mechanical energy storage and release, i.e., resilience, involving small energy dissipation upon mechanical loading and unloading, such as in resilin and elastin. These materials facilitate burst-like movements involving high stiffness and low strain and high reversibility. Synthetic hydrogels that allow highly reversible mechanical energy storage have remained a challenge, despite mimicking biological soft tissues. Here we show a synthetic concept using fixed hydrogel polymer compositions based on sequentially swollen and sequentially photopolymerized gelation steps for hierarchical networks. The sequential swellings facilitate the balance of properties between resilience and dissipation upon controlling of the chain extension. At low hierarchical levels, we show resilience with small hysteresis with increased stiffness and resilient energy storage, whereas at high hierarchical levels, a transition is shown to a dissipative and considerably reinforced state. The generality of this approach is shown using several photopolymerizable monomers.

Emerging interactive materials and soft robotics attract growing inspiration from biological functions[1,2]. The ability of a material to efficiently store and release mechanical energy to achieve rapid burst-like motions[3] is of interest for emerging soft robotics[4]. The underlying resilient biological proteins, e.g., resilin and elastin, characteristically combine reversibly high stiffness and strength at small strain upon mechanical loading and unloading, allowing to efficient store and release mechanical energy on demand with little dissipation[5].

Hydrogels are synthetic analogs of biological soft tissues. Double network hydrogels[6,7] have been widely used to achieve toughness by using highly coiled deformable networks combined with sacrificial brittle networks[8,9]. Among these, different polymers have also been incorporated up to quadruple networks[10], exploiting pre-stretching of the lower-level chains. Such mechanisms, which are fundamentally dissipative and lead to irreversibility in the stress-strain curves, are of great interest, as they allow a variety of approaches for dissipative toughening. On the other hand, biological resilient materials used for reversible mechanical energy storage and release, such as elastin and resilin[5], suggest an alternative to dissipative toughening and inspire the exploration of hydrogels with minimal mechanical energy dissipation and hysteresis. This approach has classically been demonstrated by entanglement-based mechanisms[11,12], nanoparticle reinforcement[13], slide ring gels[14], tetra-arm PEG gels[15], clay gels[16], tri-branched gels[17], polyprotein crosslinkers[18], and crosslinking density adjusted gels[19]. Approaches for synthesizing resilin have also been successful[20]. Therein, several benefits of non-hysteretic hydrogels such as absence of fatigue have been shown[14]. On the other hand, methods like phase-separation in ionic liquids[21], force-triggered chemical reactions[22], hydrogel training[23], and hybrid gels[24] yield significantly stronger gels at the cost of mechanical dissipation. However, synthetic

Department of Applied Physics, Aalto University, Espoo, Finland. ✉e-mail: mario.piedrahitabello@aalto.fi; olli.ikkala@aalto.fi

low-hysteretic hydrogels typically show low stiffness and excessively high strain, unlike their biological resilient counterparts[25].

Therefore, synthetic approaches for hydrogel reinforcements to promote high stiffness at low strains, in a reversible manner, are needed. These approaches must be capable of showing high non-dissipative energy storage and release at low strains. They would allow avenues for emerging bio-inspired soft robotics and facilitate rapid burst-like mechanical actions.

Here, we show a concept based on hierarchically swollen hydrogels, upon repeated swellings of the already formed hydrogels using a fixed swelling mixture of monomers, crosslinkers, and water with subsequent photopolymerizations up to 7 hierarchical levels N. In this multi-stage polymer synthesis, after N cycles of network polymerizations, the first networks synthesized are more stretched and therefore more stiff than the newer networks, due to increased swelling. The repeated swelling and polymerization lead to mechanical reinforcement that promotes stiffness at low strains, while still avoiding mechanical energy dissipation, i.e., allowing bioinspired resilience. However, ultimately at high N, dissipation is observed, potentially due to chain damage upon breaking of highly extended chains upon tensile deformation. Importantly, this concept can be applied to different homopolymer hydrogels facilitating hierarchically swollen polymer networks allowing hierarchical chain stretchings and nondissipative mechanical behavior. This approach also introduces the idea of self-reinforcement to hydrogel networks. This self-reinforcement concept, using a monomaterial approach to improve mechanical properties by control of the topological structure, is a strengthening paradigm that has been used before for other types of materials[26].

## Results

### Design of hierarchical swollen gels

We first explored poly(acrylamide) (PAAm) hydrogels involving up to 7 hierarchical levels of networks, wherein each hierarchical level shows characteristic chain stretching due to sequential swelling and generation-dependent chain stretching. PAAm hydrogels were selected as model materials as their hydrogelation is well known in the state of the art[27] and their swelling capacity is easily tunable via the concentration of the crosslinker. Herein, a fixed optimized photoinitiated radical polymerization protocol is sequentially used using a high acrylamide monomer concentration of 60 wt% in water, additionally incorporating a low concentration ($10^{-4}$ mol/mol) of N,N-methylene-bisacrylamide (BIS) of crosslinker vs. the acrylate monomers (for details, see Methods section). This specific acrylamide/BIS/water composition is the same as used in the subsequent swelling of the previously formed hydrogels and subsequent photo-controlled radical polymerizations.

Concretely, an initially formed PAAm hydrogel, denoted as hierarchical level N = 1 (see Fig. 1a), is next re-immersed in the above-mentioned monomer solution (acrylamide monomer/BIS crosslinker/water) which is compositionally identical to the one used in the original PAAm hydrogel synthesis. This leads to swelling of the originally formed PAAm hydrogel chains, whereupon the newly added acrylamide monomers are subsequently photopolymerized to lead to the next level hydrogel network with hierarchical level N = 2 (Fig. 1a). The newly photopolymerized chains lead to a more coiled network, which interpenetrates with the previously generated, more swollen stretched network, using a single type of polymer. The procedure can be repeated to allow higher-level hydrogel hierarchies N, presently up to N = 7 (Fig. 1a, b). These hierarchical networks resemble interpenetrating networks (IPN). However, unlike IPNs, they are composed of a single polymer type with different stretching hierarchies unlike classic IPNs[28], which are composed of polymer chains of different compositions.

During the sequential swellings and polymerizations, the previously formed polymer chain networks are increasingly swollen and stretched (Fig. 1a), thus suggesting hierarchical stretching. The intrinsically stretched polymer chains serve as scaffolds for hierarchical entanglements of the newly created chains. At each hierarchical step N, the network reacts with the monomers, leading also to the formation of bonds between the newly formed chains and the previous one. This hierarchical self-reinforcement process increases the strength and modulus due to formation of new dynamic and hierarchical entanglements while maintaining elasticity and not causing embrittlement, as is often the case for increased concentration of fixed cross-links[11]. By increasing the sequential swelling/photo-polymerization steps, the number of possible configurations of the chains diminishes as the previously formed networks undergo a high degree of extension. Therefore, the required energy to further extend the networks is expected to increase because of the loss of entropy. Thus, an entropic hierarchical self-reinforcement would seem to take place[29].

In short, the present hierarchically swollen gel consists of sequentially stretched chains because of the swelling of previously synthesized networks (Supplementary Figs. 1 and 2). This approach leads to an increase of the strength and stiffness upon increasing the hierarchical level N. The low ratio of cross-linkers to monomers ($10^{-4}$ mol/mol) guarantees formation of long chains between the crosslinks allowing stretching of the previously formed chains and coiling of the later formed chains[11]. According to 'slip-link theory', the entanglements can tune the optimal configuration and therefore transmit stress across the whole network granting the hydrogel resilience[30].

Upon increasing N, the hydrogels first become stronger and stiffer upon self-reinforcement, with an order of magnitude increase from N = 1 to 5, with only a small hysteresis (Fig. 1b and Supplementary Table 1). At each step, the polymer weight fraction is constant due to the fixed aq. swelling mixture of monomers and crosslinkers (Supplementary Fig. 3a) and the fraction of swollen polymer to non-swollen polymer increases (Supplementary Fig. 3b). Therefore, the increased mechanical properties are not due to changes in gel chemical composition but solely due to network topology, i.e., chain stretchings and entanglements. Thus, the increased mechanical properties can be coined as "self-reinforced" hydrogels. The energy storage density increases at each level of hierarchy up to N = 5, beyond which they decrease steeply probably due to excessive stretching of the first formed networks and the amount of energy stored as chain rupture may occur during mechanical extension (Fig. 1c).

### Mechanical properties of hierarchical swollen gels

In biological spring-driven resilient systems, the extension ratio of the energy storage materials has a strain limit of around 1.05-1.50.[3] Therefore, to mimic biological systems and achieve a sufficient energy storage capacity for burst-like movements, a high Young's modulus is needed (Fig. 1d and Supplementary Table 1 and 2) to achieve efficient resilience at low strains. The state-of-the-art hydrogels are unable to achieve high energy storage at such low strain values, unlike natural materials such as resilin and elastin[31]. Compared to the current state-of-the-art literature of hydrogels and biological resilient proteins, the present hierarchically swollen and entangled hydrogels show superior tensile strength and Young's modulus at small strains (Fig. 1d). From hierarchical level N = 1 to 5 the Young's modulus increases from 0.596 MPa to 1.836 MPa, the tensile strength in cyclic testing from 0.387 MPa to 3.100 MPa, and the stored energy increases from 0.487 MJm$^{-3}$ to 1.625 MJ m$^{-3}$.

However, the stress-strain behavior of the material changes fundamentally when the polymerization steps are repeated to achieve higher hierarchical degrees. Figure 1b shows that for N = 7, the stress-strain curves upon loading and unloading deviate considerably from each other, showing hysteresis, i.e., increase of energy dissipation and loss of resilience. Figure 1b shows that resilience is preserved in PAAm hydrogels up to N = 5, and mechanical energy dissipation is observed

at higher N. The PAAm hydrogel at N = 5 shows high modulus and low     hysteresis, therefore it is optimal for hysteresis-free mechanical energy

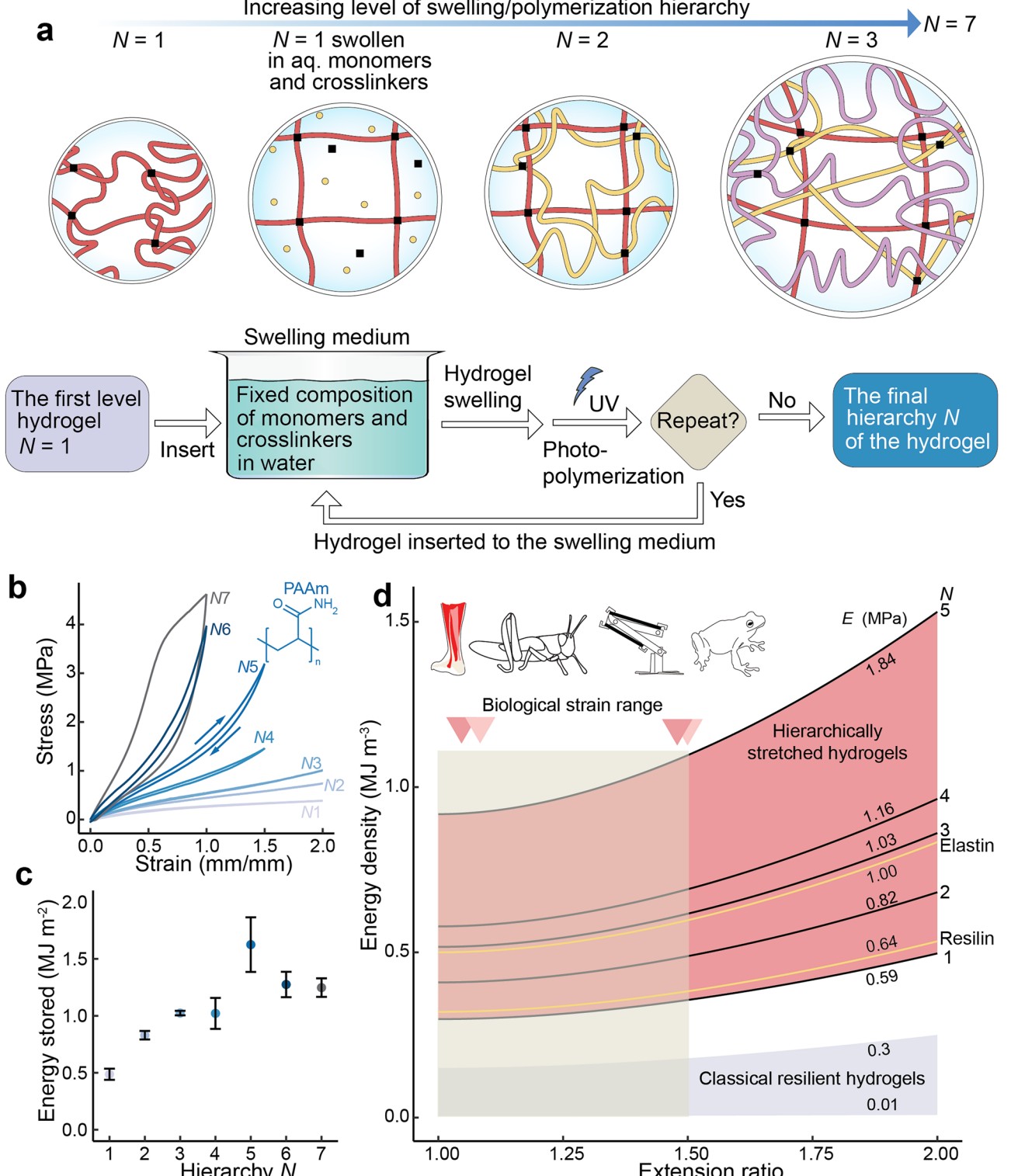

**Fig. 1 | Repeated photopolymerizations combined with hydrogel swelling steps within a fixed monomer and cross-linker aqueous composition, resulting in sequential swelling-induced hierarchical polymer chain stretching. a** Scheme for sequential swellings and polymerizations to achieve hierarchically swollen hydrogels and hierarchical stretching using a single type of monomer up to N = 7 where large hysteresis starts to occur. **b** Loading and unloading stress-strain curves of polyacrylamide (PAAm) hydrogels for hierarchical levels N = 1 to 7. They all provide self-reinforcement. Small hysteresis is observed for N = 1−5 and pronounced hysteresis for N = 6, 7. **c** Mechanical energy density stored as a function of hierarchical level N. Average of at least *n* = 3 samples is shown as point and error bar is given as standard deviation. **d** Mechanical energy density vs. strain at different stiffness values for synthetic hydrogels and biological resilient materials with biological strain range (real life examples of tendon, locust, our robot, and frog are given in triangles).

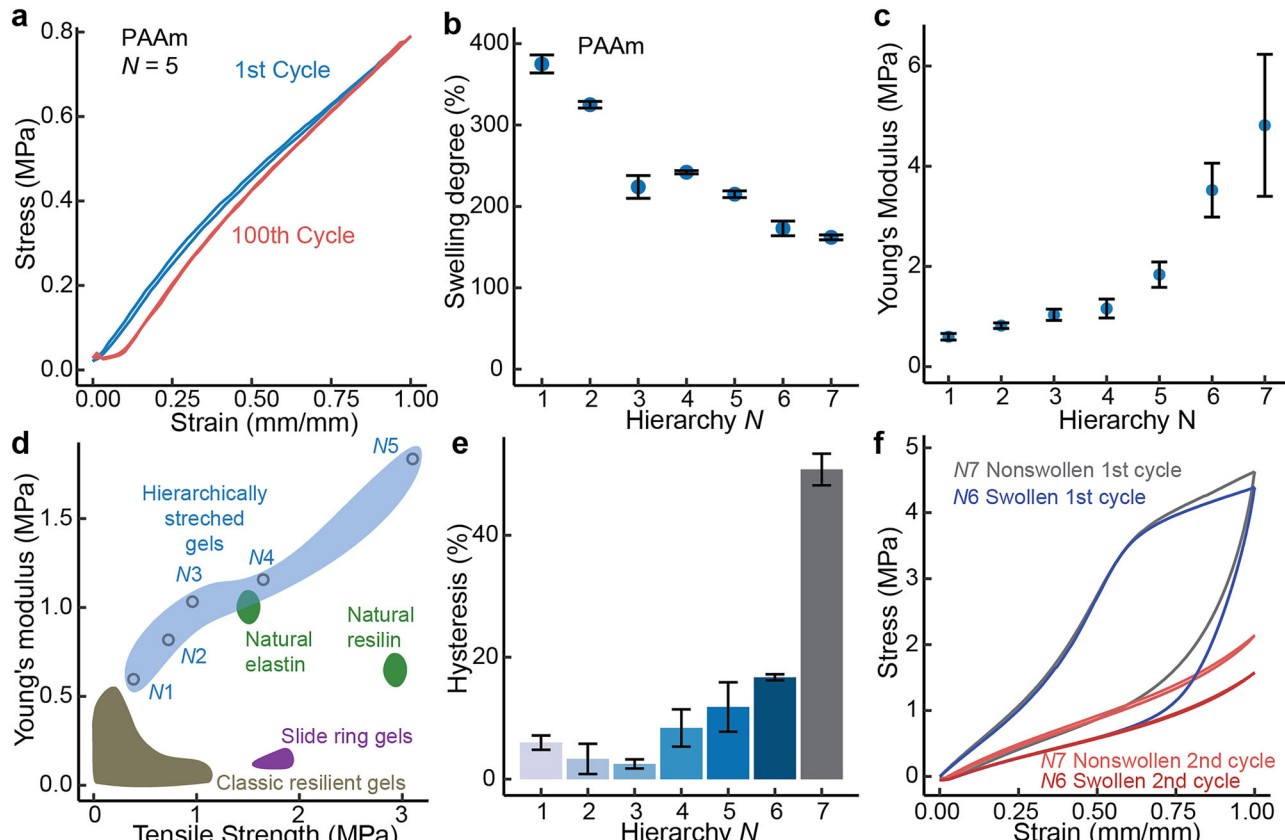

**Fig. 2 | Mechanical properties of the hierarchically stretched PAAm hydrogels.** **a** The stress-strain curves of PAAm hydrogel of N = 5 for the first tensile cycle and for the 100th cycle. **b** The stepwise change of swelling degree of hierarchical gels decreases with each subsequent increase in hierarchy N where swelling degree is gain in weight at hierarchy N in water. **c** Young's moduli as a function of N. **d** Tensile strength vs. Young's modulus in resilient hydrogels. **e** Hysteresis as a function of N. **f** Loading/unloading curves of non-swollen N = 7 and swollen N = 6 for the first and second cycles. Displayed discreet values are averages of at least $n = 3$ samples and error is given in standard deviation.

storage for PAAm hydrogels. This resilience remains constant even through 100 cycles of tensile strain of 1 mm/mm, showing minimal fatigue due to creep (Fig. 2a). The swelling degree of the gel decreases as the hierarchical level N increases (Fig. 2b and Supplementary Fig. 3c). Young's modulus increases steeply as a function of N from N = 5 up to N = 7 (Fig. 2c).

This hierarchical swelling approach allows for efficient and resilient reinforcement upon exploiting the hierarchically stretched chains, when compared to the state of the art. Breakdown of PAAm polymer chains upon tension is evidenced for N = 6, 7 due to excessive polymer stretching upon tension[32], leading to hysteresis and dissipation because of sequential swelling. This suggests an inherent limit to how many sequential reinforcement steps can be performed while retaining gel resilience. However, small hysteresis in the consecutive cycles is observed, suggesting that the next level of stretched chains are intact (Fig. 3a), only the first networks synthesized in the system have broken. However, microstructural changes in the gel are not widespread enough to be easily detectable by characterization techniques. (Supplementary Figs. 12 and S13).

## Generality of the approach

In order to prove the generality of the concept of hierarchical resilient reinforcement of hydrogels, we used different polymers to synthesize resilient reinforced hydrogels of up to N = 3. (Fig. 3b–e) For this approach we used chemically distinct polymers as a starting material, to prove the topological, rather than chemical nature of this reinforcement mechanism. We used poly(N-isopropylacrylamide) (PNIPAm), poly(oligo(ethylene glycol) methyl ether methacrylate), poly(di-

methyleneacrylamide) (PDMAA), and poly(acrylic acid) (PAAc). They all show swelling-mediated resilient self-reinforcement upon sequential radical photopolymerization (Supplementary Table 3 and Supplementary Fig. 10). The presently suggested mechanism allows self-reinforcement and low-hysteretic stress-strain behavior for hydrogels with a variety of different properties, such as thermoresponsive hydrogels (POEGMA, PNIPAm), biocompatible hydrogels (POEGMA), and pH sensitive hydrogels (PAAc).

## Design of prototype jumper based on resilient hierarchical gels

Mechanical energy storage and release in biological systems use extensively resilient proteins, e.g., elastin and resilin[1]. Locusts and other insects use an intricate, spring-like interplay of muscles, elastic tissue and chitin, i.e., latch-mediated spring actuation mechanism (LAMSA), to store mechanical energy and nondissipatively release it with a snapping motion using high mechanical energy storage at low strains (Fig. 4a)[3]. Inspired by them, we employed the resilient hierarchical N = 5 PAAm gel to store elastic energy for a demonstration of a locust-inspired jumping soft robot, fabricated via 3D printing (Fig. 4b). First, we load manually the mechanical energy of the hierarchically coiled hydrogels of a simplistic jumping soft robot. Then, we lock the structure mechanically with a strained hydrogel and release the latch via heat, i.e. thawing a PNIPAm based hydrogel acting as frozen glue, to initiate the snapping motion. With the LAMSA mechanism we show a proof of concept for burst-like mechanical energy release upon using hierarchically stretched hydrogels for actuating systems of soft robotics (Supplementary Movie 1). Actuation strength and speed have been commonly acknowledged as a major bottleneck in soft

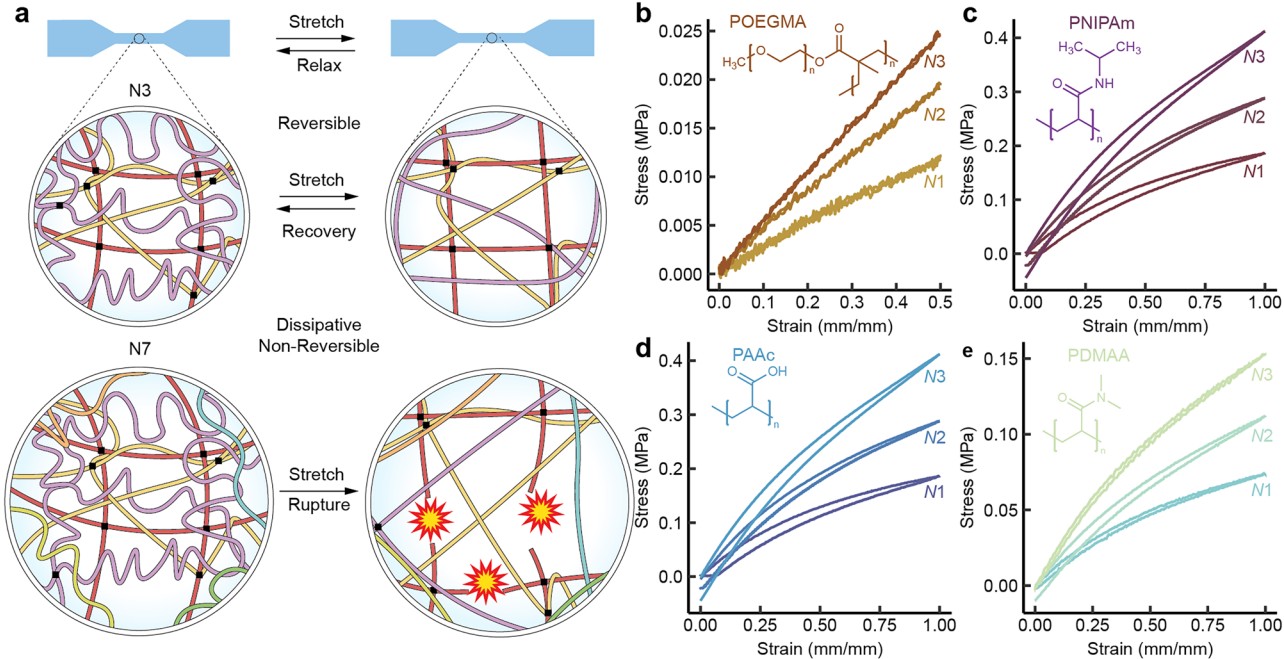

**Fig. 3 | Hierarchically stretched PAAm hydrogels under mechanical straining and generalization for other polyacrylate polymers. a** Schematic representation of the suggested stretching behavior of PAAm for a nondissipative N = 3 gel (top) and a dissipative N = 7 gel (bottom), where hysteretic behavior emerges due to the chain breaking of the highly swollen chains. **b–e** Small hysteresis at small hierarchical levels shown using several acrylate polymers.

robotics[33,34]. Latch mediated spring actuation with resilient, high stiffness, and high strength hydrogels guarantees efficient energy storage and conversion to power fast actuation which is 30% faster than that achieved with comparable classic hydrogels, resulting in a higher and faster jump (Fig. 4c, d).

## Discussion

In summary, we describe a principle for monomaterial gel strengthening which induces mechanical resilience to store and to allow efficient burst-like release of mechanical energy nondissipatively. We use sequential hydrogel swellings in fixed compositions of monomers in water and subsequent photopolymerizations to allow hierarchical polymer chain stretching. We first explored polyacrylamide hydrogels of hierarchical levels up to N = 7. Increased stiffness and ultimate strength are observed upon increased N, due to such hierarchical stretching, thus leading to the notion of self-reinforcement. At low N = 2...5, the hydrogels remain highly flexible, showing high strain, as characteristic for classic hydrogels. Significant self-reinforcement is observed and yet no hysteresis in mechanical loading and unloading is observed, showing fully reversible and resilient energy storage. By contrast, at high N = 6,7, fundamentally different mechanical properties are observed upon tension, showing drastically higher stiffness and strength and lower maximum strains. For example a high Young's modulus 5.3 MPa is observed for N = 7 vs. the low modulus of 0.7 MPa for N = 1, respectively. Noteworthily, high mechanical hysteresis is observed for N = 7, unlike that of N = 1. As the aq. monomer and cross-linker concentrations are kept constant at each hierarchical step, the reason for the increased stiffness must be related to the extensive stretching of the lower hierarchical level chain due to the sequential swellings at higher N. Also, the high hysteresis indicates nonreversible changes or damage in the network topology at high N. Given the chemically fixed cross-linked composition at all N steps, it is plausible to suggest that the dissipation follows upon chain ruptures among the most stretched chains, i.e., at low hierarchical levels. As is the case for classical double network gels[6]. The hierarchical level N = 5 appears to be the borderline case between these two fundamentally different behaviors. Therein, the

ultimate strength is increased by an order of magnitude vs. that of N = 1 and nearly 300% higher mechanical energy storage efficiency while retaining small hysteresis.

We also show hierarchical swelling as a general concept for resilient gel reinforcement using different, chemically distinct polymers. Importantly, the goal of avoiding stress-strain hysteresis for resiliency conceptually contrasts the classic double network or other previous hydrogel concepts to allow toughening wherein dissipation is herein explicitly pursued. The massively enhanced mechanical energy storage properties of the present hierarchical swelling-induced differently stretched chain gels offer great promise for applications where mechanical energy storage is crucial, such as soft robotics, specifically for jumper robots[2]. We show a proof of concept of a simplistic jumper robot powered by the mechanical energy storage of hierarchically stretched hydrogels. Our jumper prototype highlights the increase in jumping power of hierarchically stretched gels relative to classic networks. Further systematic studies to study the mechanism of self-entangled gels and the effect of various synthesis conditions on their mechanical properties will be the scope of future work.

## Methods

### Materials

Acrylamide ≥ 99%, Acrylic acid 99%, Dimethylacrylamide 99%, oligo ethylene glycol methyl ether acrylate (Mn = 500), N-isopropylacrylamide ≥ 99%, N,N'-methylenebisacrylamide ≥ 99.5, and Irgacure 2959 (98%) were purchased from Sigma Aldrich and used without any further purification.

**Synthesis of poly(acrylamide) hydrogel.** 1.5 g (0.0211 mol) acrylamide, 33 mL of 10 mg/mL (2.1 10⁻⁶ mol) $N,N'$-methylenebisacrylamide solution, and 0.3 mg (1.34 10⁻⁶ mol) 2-hydroxy-4'-(2-hydroxyethoxy)-2-methylpropiophenone (Irgacure 2959) were combined with 1 mL Milli-Q water and bubbled with nitrogen for 5 min. Then, the solution was injected into a mold consisting of two glass plates and a spacer. The mold was sealed with parafilm. Then, the sample was polymerized via irradiation for 3 h in a UV chamber (8 × 14 W, 350 nm, Rayonet, USA). After irradiation, the sample was immersed in a solution of the same

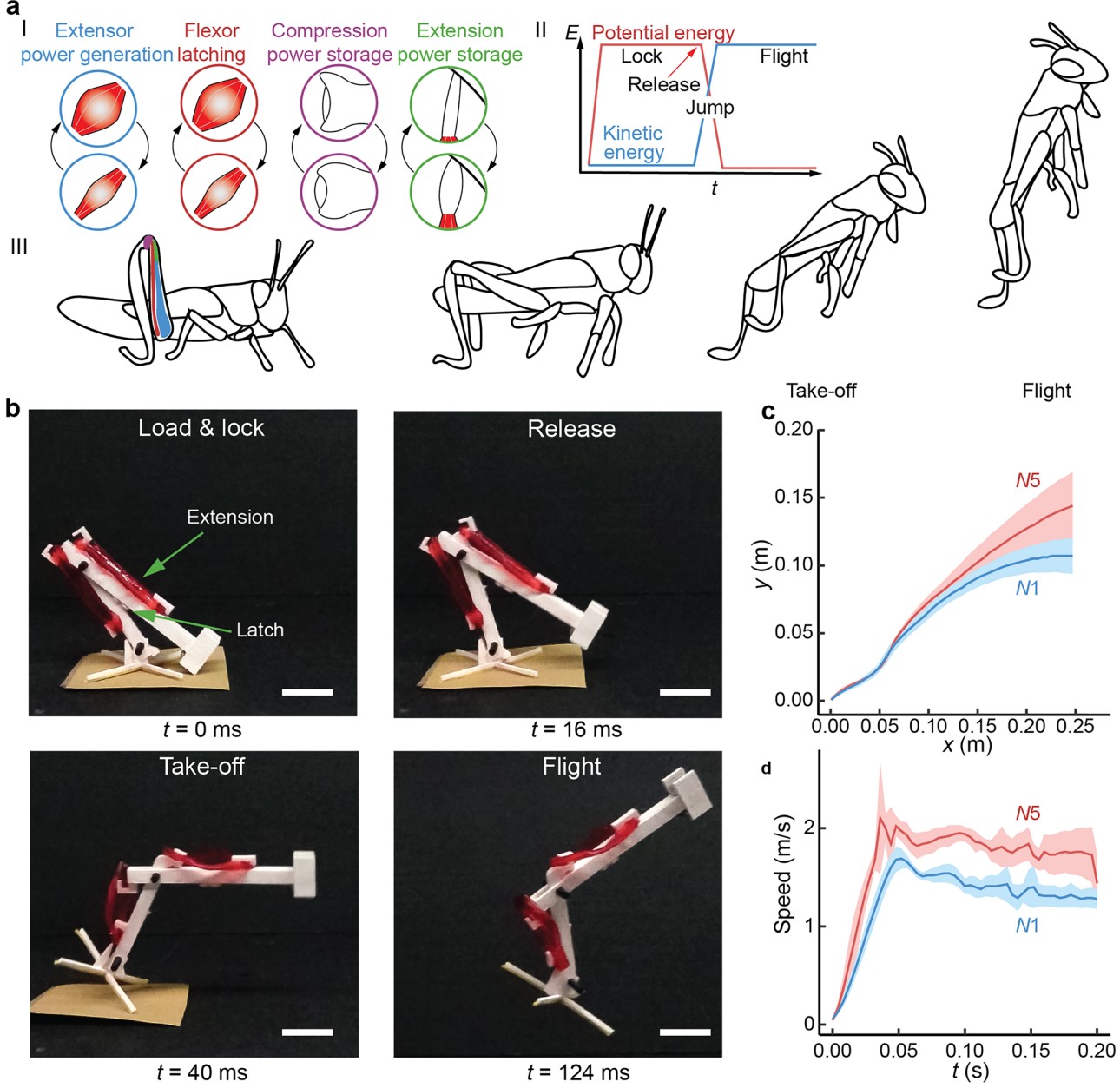

**Fig. 4 | Biological jumpers and bioinspired latch mediated spring actuation.** **a** Schematics of a locusts jump with four phases. **b** Demonstration of simplistic jumper powered using the hierarchically swollen PAAm hydrogels. Scale bar represents 30 mm. **c** The flight trajectory of the jump. **d** The jump speed of the soft jumper device. Graphs are displayed as average $n = 5$ jumps and error is given in shaded area as standard deviation.

composition overnight. Each increase in order of $N$ consists of an additional cycle of immersion and irradiation for 1 h.

**Synthesis of poly(acrylic acid) hydrogel.** 1.5 g (0.0208 mol) acrylic acid, 30 mL of 10 mg/mL ($1.95\,10^{-6}$ mol) $N,N'$-methylenebisacrylamide solution, and 1 mg ($4.46\,10^{-6}$ mol) 2-hydroxy-4'-(2-hydroxyethoxy)-2-methylpropiophenone (Irgacure 2959) were combined with 1 mL Milli-Q water and bubbled with nitrogen for 5 min. Then, the solution was injected into a mold consisting of two glass plates, a spacer and clamps. The mold was sealed with parafilm. Then, the sample was polymerized via irradiation for 3 h in a UV chamber (8 × 14 W, 350 nm, Rayonet, USA). After irradiation, the sample was immersed in a solution of the same composition overnight. Each increase in order of $N$ consists of an additional cycle of immersion and irradiation for 1 h.

**Synthesis of poly(dimethylacrylamide) hydrogel.** 1.5 g (0.0151 mol) dimethylacrylamide, 100 mL of 10 mg/mL ($6.48\,10^{-6}$ mol) $N,N'$-methylenebisacrylamide solution, and 0.3 mg ($1.34\,10^{-6}$ mol) 2-hydroxy-4'-(2-hydroxyethoxy)-2-methylpropiophenone Irgacure 2959) were combined with 1 mL Milli-Q water and bubbled with nitrogen for 5 min. Then, the solution was injected into a mold consisting of two glass plates, a spacer and clamps. The mold was sealed with parafilm. Then, the sample was polymerized via irradiation for 3 h in a UV chamber (8 × 14 W, 350 nm, Rayonet, USA). After irradiation, the sample was immersed in a solution of the same composition overnight. Each increase in order of $N$ consists of an additional cycle of immersion and irradiation for 1 h.

**Synthesis of poly(oligo ethylene glycol methyl ether acrylate) hydrogel.** 0.8 g (0.0016 mol) oligo ethylene glycol methyl ether

acrylate, 5 mL of 10 mg/mL (3.24 $10^{-7}$ mol) *N,N'*-methylenebisacrylamide solution, and 0.3 mg (1.34 $10^{-6}$ mol) 2-hydroxy-4'-(2-hydroxyethoxy)-2-methylpropiophenone (Irgacure 2959) were combined with 0.195 mL Milli-Q water and bubbled with nitrogen for 5 min. Then, the solution was injected into a mold consisting of two glass plates, a spacer and clamps. The mold was sealed with parafilm. Then, the sample was polymerized via irradiation for 3 h in a UV chamber (8 × 14 W, 350 nm, Rayonet, USA). After irradiation, the sample was immersed in a solution of the same composition overnight. Each increase in order of $N$ consists of an additional cycle of immersion and irradiatation for 1 h.

**Synthesis of Poly(N-isopropylacrylamide) hydrogel.** 0.2 g (0.00176 mol) N-isopropylacrylamide, 30 mL of 10 mg/mL (1.95 $10^{-6}$ mol) *N,N'*-methylenebisacrylamide solution, and 0.3 mg (1.34 $10^{-6}$ mol) 2-hydroxy-4'-(2-hydroxyethyl)-2-methylpropiophenone (Irgacure 2959) were combined with 0.8 mL Mili-Q water and bubbled with nitrogen for 5 min. Then, the solution was injected into a mold consisting of two glass plates, a spacer and clamps. The mold was sealed with parafilm. Then, the sample was polymerized via irradiation for 3 h in a UV chamber (8 × 14 W, 350 nm, Rayonet, USA) in an ice water bath. After irradiation, the sample was immersed in a solution of the same composition overnight. Each increase in order of $N$ consists of an additional cycle of immersion and irradiation for 1 h.

**Tensile tests.** Samples were stamped to a dumbbell shape with a width of 2 mm. The samples were glued at each end with Superglue (Loctite) to sandpaper, clamped onto an Instron 5966, and a thin layer of water was applied onto the samples. It was ensured that the humidity was at least 60% RH with a humidifier. The extension speed of the tensile tests was 2 (mm/mm) min$^{-1}$. Young's Moduli were calculated from the Strain range of 0–0.5 (mm/mm) via neohookean fit.

**Cyclic tensile tests.** Samples were stamped to a dumbbell shape with a width of 2 mm. The samples were glued at each end with Superglue (Loctite) to sandpaper. It was ensured that the humidity was at least 60% RH with a humidifier when the sample was clamped onto an onto an Instron 5966 with a box and a humidifier. The extension speed of the tensile tests was 2 (mm/mm) min$^{-1}$. Long term tests were performed at a speed of 10 (mm/mm) min$^{-1}$.

**Compression tests.** Square samples with 1 cm × 1 cm were compressed with 1 (mm/mm) min$^{-1}$ compression speed to a strain of 0.75 (mm/mm).

**Step strain tests.** Samples were stamped into a dumbbell shape with a width of 2 mm. The samples were glued at each end with Superglue (Loctite) to sandpaper and then clamped. It was ensured that the humidity was at least 60% RH with a humidifier when the sample was clamped onto an Instron 5966. The samples were extended to a strain of 0.5 (mm/mm). The extension speed of the tensile tests was 40 (mm/mm) min$^{-1}$.

**SEM analysis.** Swollen samples were shock frozen in a liquid propane bath, fractured in frozen state and then freeze dried. Samples were imaged with an Zeiss Sigma VP.

**Weight percentage determination.** Samples were weighted after polymerization. Then, swollen in Mili-Q water and weighted again. After weighing, the samples were shock frozen in liquid nitrogen and freeze dried overnight. Dried samples were weighed again. Based on the recorded weights, swelling degree, polymer weight percent, and swollen polymer weight percent were calculated.

**Fabrication of jumperBot.** The jumperbot was designed in Solidworks and printed with an Ultimaker S3 3D printer using PLA filament. Rings were cut with scissors from samples. A model of the jumper has been provided as supporting information.

**Robotics analysis.** Movies were recorded with a ZV-1 by Sony and with a framerate of 250 frames per second. The movies were analyzed via Tracker.

## Data availability

The source data generated in this study have been deposited in the figshare database under accession code https://doi.org/10.6084/m9.figshare.26363494. Additionally, all data are available from the corresponding authors upon request.

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

## Acknowledgements

We acknowledge the provision of facilities and technical support by Aalto University at OtaNano - Nanomicroscopy Center (Aalto-NMC). We acknowledge European Research Council (Advanced Grant DRIVEN No. 742829 and Dyna-Mat to O.I.), and Academy of Finland project 334433 and Center of Excellence LIBER.

## Author contributions

H.S. and M.P. envisioned the experiments, H.S. and N.H. performed the experiments, M.P., O.I., and H.S. wrote the manuscript, and all authors commented on it.

## Competing interests

The authors declare no competing interests.
