## [Transparent Peer Review file · Nature Communications]

Bioinspired nondissipative mechanical energy storage and release in hydrogels via hierarchical sequentially swollen stretched chains

Corresponding Author: Professor Olli Ikkala

Version 0:

Reviewer comments:

Reviewer #1

(Remarks to the Author)

The manuscript described a strategy of hydrogel design and fabrication leading to resilience at rather small deformation yet high modulus, mimicking those of biological elastomers. It was found that sequential polymerization of monomer in existing networks can significantly increase the modulus of obtained hydrogels. This proposed strategy is simple and of certain generality. I feel this would be very useful in developing elastic hydrogels and would suggest it to be accepted after addressing the issues listed below.

- 1) The writing needs substantial improving. Some expression and sentences are hard to understand.
- 2) cyro-SEM/TEM and small-angle scattering measurements should be performed to reveal the true microscopic structure of hydrogels.
- 3) For N=6, 7 hydrogels, the first cycling test show large hysteresis, but small in successive cyclings, this would suggest structural changes. Spectroscopic measurements, such as CD, FTIR etc should be used to follow the changes on molecular/chain level.
- 4) For most hydrogels ,resilience at low strain is guarntted, due to the limited deformation. It is suggested that the author to show the comparison of this new type hydrogel with reported strong/tough hydrogels at low strain.

Reviewer #2

(Remarks to the Author)

In this manuscript, the authors reported a novel method, which is based on hierarchically swollen hydrogels and crosslinking, to achieve high stiffness and efficient energy storage in the engineered hydrogel materials. Using such hydrogels, the authors fabricated a prototype jumper robot with excellent energy storage and release performance.

The concept is built upon but a further development of the double network strategy, and is quite interesting. However, the presentation of this manuscript and lack of technical details make it difficult to fully judge the validity of the conclusions.

- 1) Key technical parameters need to be fully defined, including swelling degree, polymer weight percent, and swollen polymer weight percent and et al.
- 2) The Young's modulus of the N1 hydrogel is 0.596 MPa, which is considerably higher than other PAAm hydrogels prepared using similar UV/chemical crosslinking, such as the one reported in Suo et al Nature, 2012, 489, 133. The authors need to explain this discrepancy.
- 3) Fig. 2b: the swelling degree of hierachical gels decreases with hierachy N. If I understand correctly, hierarchical gel N2 swells to a less degree than the gel N1. If this is the case, Fig.2b seems to contradict the description of hierachical swelling. Please explain/clarify.
- 4) Supplementary fig. 2 and other figures: it is well known that the stiffness depends on the swollen state. the authors need to clearly define the swelling degree of all the tensile measurements. Are the numbers listed in Table 1 obtained under swollen or non-swollen state? humidity 60% corresponds to swollen or non-swollen state?

5) When the authors carried out crosslinking reaction in the swollen hydrogel, the hydrogel was swollen in the mixture of monomer, initiator, crosslinker and water, what is the swelling degree of the hydrogel? my guess is the swelling ratio is different from that in pure water as reported in Fig. 2b. The authors claimed that "Such PAAm chains are more coiled than the previously formed swollen and stretched chains", this claim is based on the assumption that the newly synthesized PAAm has the same molecular weight of the elastic network strands as PAAm in the "old" network. is there any evidence for this?

6) Fig. 1a, N=1 and N=2: there are three black squares in N=1 gel, why? in N=2 gel, it appears that the newly synthesized polymer is crosslinked to the old polymer (right corner)? if this is a mistake, please correct it. If it is meant to be this way, please explain the underlying mechanism. In addition, in the first network, polymer chains are depicted as fully stretched (or close to), this is misleading. Please draw them properly.

7) Fig. 2a, comparing cycle 1 and 100, there are some clear plastic deformation. The authors should discuss this and explain its origin.

8) Reference: resilin constitutes a highly resilient material in tissues. Elvin et al reported the engineering of resilin-based biomaterials (Elvin et al Nature, 2005). This paper should be cited.

The highly entangled polymer hydrogels reported by Suo et al Science (Ref. 11) is a resilient hydrogel, the authors should properly acknowledge such prior work in engineering highly resilient materials, although this paper did not explicitly state this point but is evident in the stress-strain curves.

Reviewer #3

(Remarks to the Author)

Savolainen et al presents an interesting study on hierarchically structured hydrogel networks assembled via sequential swelling and crosslinking steps. The resulting hydrogel materials demonstrate impressive loadbearing properties in regard to maximizing mechanical energy storage (i.e. minimizing energy dissipation) during repeated loading and unloading cycles. The paper is well written, the data clearly presented and the method presented on how to assemble hierarchically structured hydrogel networks with non-dissipative properties should resonate broadly with the soft matter community. Hence I recommend publication after the authors have addressed the following points of clarification:

- On page 4, line 80, the authors mention that 'the polymer weight fraction is constant (Supplementary Fig. 3a)' with increasing hierarchical levels N1 – N5. I might have missed something, but this finding seems counterintuitive to me if the networks are becoming increasingly more densely crosslinked, which seems to be the case given the statement in the sentence right before (line 78) 'Upon increasing the final N, the hydrogels first become stronger and stiffer, with an order of magnitude increase from N = 1 to N = 5.' Can the authors please help me understand how, upon increasing hierarchical levels of network polymerization, the gels become stiffer (indicating increasing elastic chain density) without increasing the polymer weight fraction?

- In Fig. 2a, the authors showcase the 1st and 100th stress-strain cycle of an N=5 PAAm gel and concludes that this data demonstrates that 'The resilience sustains even in cyclic tests of 100 cycles of tensile strain of 1 mm/mm without fatigue, i.e., the absence of the Mullins effect (Fig. 2a).' (page 7, line 112-113). While I agree with this conclusion, there is however a difference in the stress-strain curve of the 100th cycle with a softening appearing in the first 10% of strain. Can the authors clarify this effect in the 100th cycle further?

- Finally, given the strong inspiration and referencing by the authors to biological materials such as elastin and resilin, I was a bit surprised that I did not see any discussion of (or comparison to) the entropic origin of the perfect rubber-like properties of these biological materials. Seminal work by people such as Torkel Weis-Fogh demonstrated elegantly how these biological rubbers possess their impressive non-dissipative elastic properties thanks to the perfect entropic behavior of their protein hydrogel networks, by applying classic thermo-mechanical tests. I don't think I recall the authors discussing the possible role of entropy in the similarly non-dissipative elastic behavior of their hierarchically assembled hydrogel networks. I am not saying that the authors have to do similar classical thermo-mechanical investigations to explore the possible contribution of entropy to the ideal rubber behavior observed in these new materials, but it would no doubt add a much deeper level of understanding if such studies were possible. If the authors are interested in exploring this further, I recommend using the seminal paper by Torkel Weis-Fogh listed below as a guide, since the thermo-mechanical analysis applied in this paper would allow the authors to elucidate whether the impressive resilin-like properties of their new hierarchically entangled networks likewise stem from an entropic origin.

https://www.sciencedirect.com/science/article/pii/S0022283661800181?ref=cra_js_challenge&fr=RR-1

Version 1:

Reviewer comments:

Reviewer #1

(Remarks to the Author)

I am still concerned with the structural evolution of hydrogels N=6 and 7 during the first cycle. As shown in Fig 2f, huge hysteresis was observed, while the authors did not find any structural changes before and after the cyclic tension, either on

molecular level or above. This is hard to believe. I feel if the structural origin for this mechanical property can be identified, approaches to avoid this "defect" in first cycle may be developed, making these hydrogels truly "resilient", rather than "conditionally resilient". For this reason, I would like to urge the authors to tackle this puzzle rather than leave it behind.

Reviewer #2

(Remarks to the Author)

The authors have adequately addressed my comments. I recommend its publication.

Reviewer #3

(Remarks to the Author)

The authors have addressed my questions and I believe the paper is now ready for publication.

Re: Manuscript “*Bioinspired non-dissipative mechanical energy storage and release in hydrogels via hierarchically stretched chains by sequential swelling*”, by Henri Savolainen, Negar Hosseiniyan, Mario Piedrahita-Bello, and Olli Ikkala, Nature Communications, NCOMMS-24-45609-T

Responses for the Reviewers’ questions and criticisms

We thank for the reviewers for their constructive criticisms and helpful remarks (in italics). Hereby we will provide our answers (in bold):

Reviewer #1 (Remarks to the Author):

The manuscript described a strategy of hydrogel design and fabrication leading to resilience at rather small deformation yet high modulus, mimicking those of biological elastomers. It was found that sequential polymerization of monomer in existing networks can significantly increase the modulus of obtained hydrogels. This proposed strategy is simple and of certain generality. I feel this would be very useful in developing elastic hydrogels and would suggest it to be accepted after addressing the issues listed below.

1) The writing needs substantial improving. Some expression and sentences are hard to understand.

Our response: We thank for the positive comments and we appreciate for pointing the generality of our approach. We also thank for the remarks and criticisms, wherein we extensively revised the text of the manuscript to allow improved clarity, we also suggest to revise the title as “Bioinspired non-dissipative mechanical energy storage and release in hydrogels by hierarchically stretched chains by sequential swellings” to be more concrete. Therein, we attach also a revised version where the text changes have been highlighted.

2) cyro-SEM/TEM and small-angle scattering measurements should be performed to reveal the true microscopic structure of hydrogels.

Our response: We thank for the suggestions to include new relevant characterizations. In fact, already in the already submitted version as Supplementary Figures 4, we show the cryo-SEM/freeze-dried micrographs of the hydrogels. As a state of the art background, several recent articles indicate that freeze-dried SEM of hydrogels can be feasible for morphological characterization (M. Hua et al, *Nature*, 2021, 590, 594; R. Zhu et al, *Nat. Commun.* 2024, 15, 1344; G. Zhang et al, *Sci. Adv.*, 2023, 9, eadh7742). Herein, to allow even potential improvements in the fidelity of the characterisation, we exploit first rapid freeze drying in liquid propane, followed by subsequent further cooling to liquid nitrogen. In our earlier studies this approach turned a feasible way to largely suppress the

unwanted extent of aggregations typically observed upon the classic direct freezing using liquid nitrogen, which is known to suffer from the Leidenfrost effect, as we have previously shown (J. Korhonen et al, *ACS Nano*, 5, 1967, 2011). Thus, our already present cryo-SEM micrographs qualitatively indicated more “crowding” of the networks upon higher hierarchies N , whereas more detailed quantitative interpretations seem challenging.

Cryo-TEM has been successful to show structures in colloidal gels, in general. For molecular gels, like the present one, the resolution and contrast provide generic problems, typically requiring labelling by, e.g., suitable nanoparticles. Therefore we did not find such an approach sufficiently feasible/informative/attractive to warrant the considerable further efforts involved.

As asked by the reviewer, new SAXS measurements have now been attached as Supplementary Figures 13, also attached herein. However, they just suggest general behaviors that are expected for hydrogels. They suggest an overall scaling relation $I \propto q^{-n}$ wherein the scaling exponent n gives qualitative information. In this case, at small N , the n is typically close to 2 whereas for increased N , n starts to approach 3. Qualitatively this suggests that upon increasing hierarchy, the extent for “fractal” texture is increased. More quantitative interpretation is challenging.

Figure. SAXS data to show structural evolution upon hierarchy N .

3) For $N=6, 7$ hydrogels, the first cycling test show large hysteresis, but small in successive cyclings, this would suggest structural changes. Spectroscopic measurements, such as CD, FTIR etc should be used to follow the changes on molecular/chain level.

Our response: Since PAAm-based hydrogels are amorphous and acrylamide is not optically active, CD does not allow concrete avenues to characterize the gel. FTIR spectra were performed and were already in Supplementary Figures 12-13. They indicated no specific changes to the gel chemical composition before and after stretching. These results are now mentioned in the main text on page 7, written as “whereas we do not see any significant changes in the microstructure (Supplementary Figures 12 & 13).” SAXS was performed to investigate the structural changes of the hydrogel before and after stretching. There were no specific changes between the gels before and after swellings as seen in Supplementary Figures 12-13.

For $N = 6,7$ we can infer from the mechanical testing data that the Mullins effect takes place, a very common effect in elastomers and hydrogels (R. Webber, et al *Macromolecules* 2007, 40, 2919). The network ruptures at many different places upon mechanical stressing, causing the hysteresis in the retraction and subsequent weakening of the hydrogel in the next cycle. We have now included this in the main text under page 7 for clarity. “Breakdowns of polymer chains at $N = 6,7$ due to excessive polymer stretching upon tension is suggested³², leading to hysteresis and dissipation but smaller hysteresis in the consecutive cycles”.

4) For most hydrogels ,resilience at low strain is guaranttd, due to the limited deformation. It is suggested that the author to show the comparison of this new type hydrogel with reported strong/tough hydrogels at low strain.

Our response: We have looked through the literature of high strength hydrogels and have not found examples of resilience in them at the high strains shown in this work. This is due to their inherent dissipation during deformation which causes them to have high toughness. Even in the region where the hydrogels reorientate, network rupture can happen. A strain of 1 (mm/mm) as we show in this work is well above to this region.

Reviewer #2

In this manuscript, the authors reported a novel method, which is based on hierarchically swollen hydrogels and crosslinking, to achieve high stiffness and efficient energy storage in the engineered hydrogel materials. Using such hydrogels, the authors fabricated a prototype jumper robot with excellent energy storage and release performance.

The concept is built upon but a further development of the double network strategy, and is quite interesting. However, the presentation of this manuscript and lack of technical details make it difficult to fully judge the validity of the conclusions.

1) Key technical parameters need to be fully defined, including swelling degree, polymer weight percent, and swollen polymer weight percent and et al.

Our response: We appreciate the criticisms and have now explained in depth the denotions. Herein, we further explain them, also related to Supplementary Figures 3.

“The swelling degree is defined as follows

$$\rho = \frac{\text{weight}_{\text{swollen}}}{\text{weight}_{\text{pristine}}}$$

The polymer weight % is defined as follows

$$\rho = \frac{\text{weight}_{\text{dry}}}{\text{weight}_{\text{pristine}}}$$

The swollen polymer weight % is defined as follows

$$\rho = \frac{\text{weight}_{\text{dry}}}{\text{weight}_{\text{swollen}}}$$

2) The Young's modulus of the N1 hydrogel is 0.596 MPa, which is considerably higher than other PAAm hydrogels prepared using similar UV/chemical crosslinking, such as the one reported in Suo et al Nature, 2012, 489, 133. The authors need to explain this discrepancy.

Our response: The discrepancy can be explained by the fact that the synthesis procedure is completely different (vs. Suo et al, Nature, 2012, 489, 133), thus also suggesting the importance of the present processing. For example, we use a different light source (8x 14 W, 350 nm, Rayonet, USA vs. 8 ea, Sankyo Denki, F8T5BL), performing degassing with nitrogen of the sample, and using only glass plates as the mold. The presence of oxygen in the water can also inhibit the radical polymerization. The given hydrogels in Suo’s publication are swollen overnight in water while our sample in the main text is non-swollen at N1. Supplementary Figure 2a shows a PAAm hydrogel which is swollen with only one network. The mechanical properties of this gel are similar to those of the hydrogel reported by Suo et al, Nature, 2012, 489, 133.

3) Fig. 2b: the swelling degree of hierachical gels decreases with hierachy N. If I understand correctly, hierarchical gel N2 swells to a less degree than the gel N1. If this is the case, Fig.2b seems to contradict the description of hierachical swelling. Please explain/clarify.

Our response: We have now changed the sentence for improved clarity.

“The stepwise change of swelling degree of hierarchical gels decreases with each subsequent increase in hierarchy N where swelling degree is gain in weight at hierarchy N in water.”

4) Supplementary Figure 2 and other figures: it is well known that the stiffness depends on the swollen state. the authors need to clearly define the swelling degree of all the tensile measurements. Are the numbers listed in Table 1 obtained under swollen or non-swollen state? humidity 60% corresponds to swollen or non-swollen state?

Our response: We have now clarified the experimental conditions in Supplementary Table 1 to avoid confusion.

“Mechanical properties of non-swollen PAAm based hydrogels. Measurements were done at 60% air humidity.”

5) When the authors carried out crosslinking reaction in the swollen hydrogel, the hydrogel was swollen in the mixture of monomer, initiator, crosslinker and water, what is the swelling degree of the hydrogel? my guess is the swelling ratio is different from that in pure water as reported in Fig. 2b. The authors claimed that "Such PAAm chains are more coiled than the previously formed swollen and stretched chains", this claim is based on the assumption that the newly synthesized PAAm has the same molecular weight of the elastic network strands as PAAm in the "old" network. is there any evidence for this?

Our response: We have now added swelling measurements of the hydrogel swollen in different media as Supplementary Figure 3c and added the following sentence to the text.

“The swelling decreases as the final hierarchical level N increases in water and monomer solution (Fig. 2b & Supplementary Fig. 3c)”

6) Fig. 1a, $N=1$ and $N=2$: there are three black squares in $N=1$ gel, why? in $N=2$ gel, it appears that the newly synthesized polymer is crosslinked to the old polymer (right corner)? if this is a mistake, please correct it. If it is meant to be this way, please explain the underlying mechanism. In addition, in the first network, polymer chains are depicted as fully stretched (or close to), this is misleading. Please draw them properly.

Our response: Jian Ping Gong’s previous work (T. Nakajima et al, *Macromolecules* 2009, 42, 2184) on double network hydrogels suggests that there are still open polymerization sites after the first polymerization which can react with the second formed network. The black squares are the crosslinker which is required for this mechanism to be possible, as seen in Supplementary Figure 8. We added the following section to the paper to clarify the drawing on page 4.

“During the next steps, the network reacts with the monomers to form an interpenetrating network (IPN), as for clearly distinct network additional treatment would be required.

Fig. 1a is meant to be a pictorial representation of the mechanism of stretching chains rather than an accurate depiction. This has now been made clear in the description on page 6. This representation is meant to showcase the hysteresis and therefore rupture of gels of hierarchical level 7 and swollen gel of hierarchical level 6.

“Scheme for sequential swellings and polymerizations to achieve hierarchically swollen hydrogels and hierarchical stretching using a single type of monomer *with the last network showing the effect of this process on the network reaching up to $N = 7$ where large hysteresis starts to occur.*”

7) Fig. 2a, comparing cycle 1 and 100, there are some clear plastic deformation. The authors should discuss this and explain its origin.

Our response: Indeed, we can infer that there is slow creep as the plastic deformation as resilient hydrogels are not immune to creep phenomena. We corrected the sentence on page 7 to:

‘The resilience sustains even in cyclic tests of 100 cycles of tensile strain of 1 mm/mm shows minimal fatigue due to creep. (Fig. 2a).

8) Reference: resilin constitutes a highly resilient material in tissues. Elvin et al reported the engineering of resilin-based biomaterials (Elvin et al Nature, 2005). This paper should be cited.

Our response: We thank the reviewer for the suggestion and have added the citation in the intro section as this segment.

“Approaches of synthesizing resilin have been successful.⁴” (C. Elvin, et al. Nature 2005, 437, 999).

The highly entangled polymer hydrogels reported by Suo et al Science (Ref. 11) is a resilient hydrogel, the authors should properly acknowledge such prior work in engineering highly resilient materials, although this paper did not explicitly state this point but is evident in the stress-strain curves.

Our response: We thank the reviewer for the suggestion and we now mention explicitly the paper in the section where we discuss mechanisms for entanglements on page 2 shown in bold.

“On the other hand, hydrogel reinforcement with minimal mechanical energy dissipation and hysteresis have been demonstrated by entanglement-based mechanisms^{11,12}, nanoparticle reinforcement¹³, slide ring gels¹⁴, tetra-arm PEG gels¹⁵, clay gels¹⁶, tri-branched gels¹⁷, polyprotein crosslinkers¹⁸, and crosslinking density adjusted gels¹⁹” .

Reviewer #3

Savolainen et al presents an interesting study on hierarchically structured hydrogel networks assembled via sequential swelling and crosslinking steps. The resulting hydrogel materials demonstrate impressive loadbearing properties in regard to maximizing mechanical energy storage (i.e. minimizing energy dissipation) during repeated loading and unloading cycles. The paper is well written, the data clearly presented and the method presented on how to assemble hierarchically structured hydrogel networks with non-dissipative properties should resonate broadly with the soft matter community. Hence I recommend publication after the authors have addressed the following points of clarification:

- On page 4, line 80, the authors mention that ‘the polymer weight fraction is constant (Supplementary Fig. 3a)’ with increasing hierarchical levels N1 – N5. I might have missed something, but this finding seems counterintuitive to me if the networks are becoming increasingly more densely crosslinked, which seems to be the case given the statement in the sentence right before (line 78) ‘Upon increasing the final N, the hydrogels first become stronger and stiffer, with an order of magnitude increase from N = 1 to N = 5.’ Can the authors please help me understand how, upon increasing hierarchical levels of network polymerization, the gels become stiffer (indicating increasing elastic chain density) without increasing the polymer weight fraction?

Our response: We assume that swelling decreases the volumetric density of chains because the network extends and takes up more solvent but at the same time the network takes up monomers to increase the polymer content after the polymerization. The stiffening for each hierarchical level is due to the increase of the density of polymer chains, entanglements between them, and entropic contribution. We added a text provided in the last answer to the reviewer as further explanation for the increase of stiffness of the network on page 4.

“ Each subsequent step causes the sample to swell more, extending each network in a different degree step by step. By extending the chain due to swelling/expansion of the network, the number of possible configuration of the chains diminishes where the first network has the highest degree of extension. As a consequence, the required energy to further extend the networks, increases because of the loss of entropy, therefore entropic self-reinforcement taking place.”

- In Fig. 2a, the authors showcase the 1st and 100th stress-strain cycle of an N=5 PAAm gel and concludes that this data demonstrates that ‘The resilience sustains even in cyclic tests of 100 cycles of tensile strain of 1 mm/mm without fatigue, i.e., the absence of the Mullins effect (Fig. 2a).’ (page 7, line 112-113). While I agree with this conclusion, there is however a difference in the stress-strain curve of the 100th cycle with a softening appearing in the first 10% of strain. Can the authors clarify this effect in the 100th cycle further?

Our response: Indeed, we assume that there is slow creep as the plastic deformation remains upon several cycles. Herein, we should observed a very small creep which causes slow lengthening of the sample due to a small amount of chain rupture at each extension cycle, therefore no noticeable resistance to 10% of strain. Therefore, we corrected the sentence on page 7 to:

‘The resilience sustains even in cyclic tests of 100 cycles of tensile strain of 1 mm/mm shows minimal fatigue due to creep (Fig. 2a).’

- Finally, given the strong inspiration and referencing by the authors to biological materials such as elastin and resilin, I was a bit surprised that I did not see any discussion of (or comparison to) the entropic origin of the perfect rubber-like properties of these biological materials. Seminal work by people such as Torkel Weis-Fogh demonstrated elegantly how these biological rubbers possess their impressive non-dissipative elastic properties thanks to the perfect entropic behavior of their protein hydrogel networks, by applying classic thermo-mechanical tests. I don’t think I recall the authors discussing the possible role of entropy in the similarly non-dissipative elastic behavior of their hierarchically assembled hydrogel networks. I am not saying that the authors have to do similar classical thermo-mechanical investigations to explore the possible contribution of entropy to the ideal rubber behavior observed in these new materials, but it would no doubt add a much deeper level of understanding if such studies were possible. If the authors are interested in exploring this further, I recommend using the seminal paper by Torkel Weis-Fogh listed below as a guide, since the thermo-mechanical analysis applied in this paper would allow the authors to elucidate whether the impressive resilin-like properties of their new hierarchically entangled networks likewise stem from an entropic origin.

Our response: We thank the author for the insightful comment and we will add, therefore, a thermodynamic perspective. While there are many phenomena occurring in a bulk hydrogel such as entanglements, reconfiguration, and reptation, we agree that the thermodynamic perspective has been neglected in our analysis even though it is important, therefore we added the following paragraph to our explanation of the system on page 4.

We have added a text “ Each subsequent step causes the sample to swell more, extending each network in a different degree step by step. By extending the chain

due to swelling/expansion of the network, the number of possible configuration of the chains diminishes where the first network has the highest degree of extension. As a consequence, the required energy to further extend the networks, increases because of the loss of entropy, therefore entropic self-reinforcement taking place.”

Re: Manuscript “*Bioinspired non-dissipative mechanical energy storage and release in hydrogels via hierarchically stretched chains by sequential swelling*”, by Henri Savolainen, Negar Hosseiniyan, Mario Piedrahita-Bello, and Olli Ikkala, Nature Communications, NCOMMS-24-45609-T

Responses for the Reviewers’ questions and criticisms

We thank the reviewer for the constructive criticism and helpful remarks (in italics). Hereby we will provide our answers (in bold):

Reviewer #1 (Remarks to the Author):

I am still concerned with the structural evolution of hydrogels N=6 and 7 during the first cycle. As shown in Fig 2f, huge hysteresis was observed, while the authors did not find any structural changes before and after the cyclic tension, either on molecular level or above. This is hard to believe. I feel if the structural origin for this mechanical property can be identified, approaches to avoid this "defect" in first cycle may be developed, making these hydrogels truly "resilient", rather than "conditionally resilient". For this reason, I would like to urge the authors to tackle this puzzle rather than leave it behind.

Our response: We thank for the reviewer for his/her criticism and suggesting us to consider more deeply the findings.

Altogether, the system is highly complex, involving chains of several swelling/stretching states. In the state of the art, detailed mechanistic understanding even of single nonswollen/swollen hydrogel is presently a challenge, not to mention 7 hierarchical swellings of the present case. Complete understanding of the hierarchies is clearly beyond reach at present in simulations or theory.

Still, qualitative suggestions can be made, explained herein, also highlighted in the main text in blue.

The state of the art of double network, nonresilient hydrogels largely describes mechanical energy dissipation to allow toughening upon combining a minor fraction of “harder” brittle reinforcing and sacrificial polymer networks with a major fraction of deformable polymer networks. The breaking of such harder networks allows toughening under mechanical tests to allow energy dissipation and toughness at the cost of the loss of resilience, via chain breaking. This has even been verified using Fenton’s reactions (Science 2019 363 504).

The reviewer’s comments suggested us to reconsider and deepen the message and findings. Based on the suggestion, we provide revisions in the text, highlighted in blue. The question herein is whether instead of classically using two different aq. networks, i.e., hard and sacrificial and soft deformable ones, could one use just one polymer type network upon its hierarchical swelling and extended conformations to allow reinforcement, while retaining resilience.

So, the findings of this manuscript can be summarized as follows:

1. At low $N \leq 4$. The system is in the slightly reinforced resilient state whereupon the reinforcing swollen chains at such lower N are only mildly stretched and are not ruptured in tension. It suggests that the coiled structure of the chains is preserved. This case allows resilience and dynamics.

2. At high $N = 6,7$ The system is in the extensively reinforced state whereupon the lower level chains are highly stretched allowing high reinforcements but become vulnerable to rupture upon tension. At this level of hierarchical entanglement, the system behaves like a classical double network, with the brittle “sacrificial” network being the initial, highly swollen and tense networks.

3. The limiting case $N = 5$ Showing meaningful reinforcement and retaining resilience.

So, the findings also emphasize hierarchy-driven transition from resilient small dissipation with fully stretchable networks to classical pseudo-double network energy dissipative system with sacrificial highly extended brittle networks to allow fracture energy dissipation via bond cleavage.

Therein, the transition only relates to chain conformation, not their chemical composition.

Finally this hypothesis can be appreciated based on the data:

Fig. 2f shows the transition from resilient nondissipative to dissipative tough state

Fig. 2c shows the transition from the mildly reinforced to the strongly reinforced state

Fig. 2e shows that upon transition to the strongly reinforced state, wherein the hysteresis is increased.

Fig. 2f suggests a Mullins-like hysteresis and chain damage upon tension in $N = 7$, whereas the absence in $N = 5$.

Finally, we tried also to explore the creation of radicals upon chain ruptures using Fenton reactions. This turned unsuccessful, probably due to the small fraction of chain ruptures. Also, exploring the MW of polymers after mechanical testing is impossible due to the cross-linking.

We have extensively revised the Abstract and part of the Discussion highlighted by blue color, to ensure better clarity on this mechanism, as required by the reviewer.

Minor editorial revisions are additionally highlighted in yellow color to improve fluidity of the text for editorial purposes.